# Single-Cell mRNA Analysis for the Identification of Molecular Pathways of IRF1 in HER2+ Breast Cancer

**DOI:** 10.3390/cells14161246

**Published:** 2025-08-13

**Authors:** Laura Vilardo, Paride Pelucchi, Antonia Brindisi, Edoardo Abeni, Eleonora Piscitelli, Ettore Mosca, Giovanni Bertalot, Mira Palizban, Theodoros Karnavas, Angelos D. Gritzapis, Ioannis Misitzis, Martin Götte, Ileana Zucchi, Rolland Reinbold

**Affiliations:** 1Institute of Biomedical Technologies, National Research Council, 20054 Milano, Italy; laura.vilardo@itb.cnr.it (L.V.); paride.pelucchi@itb.cnr.it (P.P.); antonia.brindisi@itb.cnr.it (A.B.); edoardo.abeni@gmail.com (E.A.); eleonora.piscitelli@itb.cnr.it (E.P.); ettore.mosca@itb.cnr.it (E.M.); 2Unita’ Operativa Multizonale di Anatomia Patologica, APSS and Centre for Medical Sciences—CISMed, University of Trento, 38122 Trento, Italy; giovanni.bertalot@apss.tn.it; 3Department of Gynecology and Obstetrics, University Hospital Muenster, D11, 48149 Muenster, Germany; mirapalizban@hotmail.com (M.P.); martin.goette@ukmuenster.de (M.G.); 4Department of Biology, Touro University, New York, NY 10023, USA; th.karnavas@gmail.com; 5Cancer Immunology and Immunotherapy Center, Agios Savas Cancer Hospital, 11522 Athens, Greece; agkriitzapis@agsavvas-hosp.gr; 6Athens Medical Center, Psychiko Clinic, 11525 Athens, Greece; imissitzis@gmail.com; 7Associazione Fondazione Renato Dulbecco, 20138 Milano, Italy

**Keywords:** single-cell transcript analysis, *HER2*+ breast cancer, *IRF1*, *RNASET2*, *TMEM230*, rheumatoid arthritis, *SDC2*, endoplasmic reticulum, *TP53*, *interferons*

## Abstract

Clonally established tumor cell lines often do not recapitulate the behavior of cells in tumors. The sequencing of a whole tumor tissue may not uncover transcriptome profiles induced by the interactions of all different cell types within a tumor. Interferons for instance have a vast number of binding sites in their target genes. Access to the DNA binding sites is determined by the epigenomic state of each different cell type within a tumor mass. To understand how genes such as interferons appear to have both tumor-promoting and tumor-inhibiting functions, single-cell transcript analysis was performed in the breast cancer tissue of *HER2*+ (epidermal growth factor receptor 2) patients. We identified that potential antagonistic oncogenic activities of cells can be due to diverse expression patterns of genes with pleiotropic functions. Molecular pathways both known and novel were identified and were similar with those previously identified for patients with rheumatoid arthritis. Our study demonstrates the efficacy in using single-cell transcript analysis to gain insight into genes with apparent contradictory or paradoxical roles in oncogenesis.

## 1. Introduction

We previously demonstrated that *TMEM230*, a transmembrane protein, is a master regulator of the endoplasmic reticulum (ER), controlling various functions, including the initial steps of glycosylation in glycoprotein and proteoglycan formation, as well as their intracellular trafficking and secretion [1]. *TMEM230*-regulated glycoproteins and proteoglycans, as we have previously shown for *RNASET2* and syndecans, regulate cell-to-cell and cell-to-extracellular matrix interactions and cell “defense” programs, respectively. These interactions were necessary for the evolutionary development of the immune system’s response to infection and cellular damage, and of tissue and extracellular matrix remodeling in wound healing and cancer [2]. As a transmembrane protein of the ER, *TMEM230* is also essential in the intracellular trafficking and secretion of glycosylated factors in endosomes, phagosomes, lysosomes, and exosomes. *TMEM230*, syndecans, and *RNASET2* were previously identified in molecular pathways associated with interferons (IFNs) [3]. IFNs are cytokine-signaling factors, initially recognized as being synthesized in response to infections but now known to have roles in autoimmunity and in paradoxically promoting and inhibiting cancer [3,4,5,6,7,8,9,10,11]. The expression of IFNs is regulated by transcription interferon regulatory factors (IRFs), such as IRF1. Like IFNs, *RNASET2*, syndecans, and *TMEM230* appear to have both pro- and anti-cancer activities, depending on their levels of expression in different cells of the tumor mass [3]. Interpretations as to why pleiotropic-acting genes may have cancer -promoting and -inhibiting activities have previously suggested this to be due to a large number of mutations, alternative splicing isoforms, or binding sites in target genes [8,12,13,14,15,16,17,18]. In this study, we identified expression patterns of *IRF1*, *TMEM230*, *RNASET2*, and syndecan 2 (*SDC2*) in different cell types of breast cancer patients using single-cell mRNA transcript analyses. We propose that uncovering their associated pathways in different cell types within the same tumor of a patient may provide insight into how pleiotropic-acting genes contribute both tumor-inhibiting and -promoting functions.

## 2. Materials and Methods

### 2.1. Dominant Acting IRF1+I4 Isoform Lentiviral Construct Generation

IRF-1+I4 isoform was cloned under a CMV promoter with the expression cassette for the green fluorescent reporter gene (*copGFP*) by cloning a PCR product into the pCDH-CMV-MCS-EF1-copGFP vector (System Bioscience, Embarcadero Way, Palo Alto, CA, USA). Additionally, the *IRF1* WT transcript in frame with the *GFP* gene and the *GFP* gene alone (used as controls) were also cloned into the lentiviral constructs and transduced like *IRF1*+*I4* into primary fibroblast cells of breasts from normal patients who had undergone mammoplasty reduction (Appendix A). *IRF1*+*I4* was amplified by primers designed in *IRF1* gene intron 4, Fw 5′-AGGGAGGGTAGAAGGAGGTCA-3′, and exon 6 Rev 5′-TGCTGAGTCCATCAGAGAAGGTAT-3′.

### 2.2. Lentiviral Transduction

For lentiviral transduction, 1 × 10^5^ cells/well were seeded in 24-well tissue culture plates and infected the following day with lentiviruses. All infections were performed for 16 h in the presence of 8 µg/mL polybrene transfection reagent (Sigma-Aldrich, St. Louis, MO, USA). High-titer lentiviruses were generated by transient co-transfection of 293 T cells with a three-plasmid combination as follows: one one 15 cm plate containing 6 × 10^6^ 293 T cells was co-transfected using Lipofectamine™ 2000 (Invitrogen™ Thermo Fisher Scientific, Waltham, MA, USA) according to the manufacturer’s protocol, with 33.25 µg lentiviral vector, 25 µg psPAX2, and 8.25 µg pMD2.G. Supernatants were collected every 24 h between 36 and 72 h after transfection, pulled together, concentrated via ultracentrifugation, and frozen at −80 °C.

### 2.3. Fluorescent Activated Flow Cytometry of GFP Expressing Cells

Fibroblast cells from breast tissue from normal patients that were transduced with *IRF1*+*I4* displayed a decrease in cell number in tissue culture conditions compared to control GFP transduced cells, or control cells expressing wildtype IRF1 as determined by fluorescent activated cell cytometry analysis (Appendix A). Human fibroblast (ranging from 1 × 10^4^ to 3 × 10^4^ cells in 100 μL PBS Buffer) were used for flow cytometric analysis with FACS CantoTM II (BD Biosciences, Franklin Lakes, NJ, USA).

### 2.4. Primary Cell Culture Conditions

Human fibroblasts samples were obtained from normal breast from mammoplastic reduction procedures as previously described [4]. Cells were cultured in 0.1% gelatin-coated plates (Sigma Aldrich, St. Louis, MO, USA) in Ham’s F12/ DMEM-GlutaMAX (1:1) (DMEM, Euroclone, ECB7501L; Pero, Milan, Italy) containing 10% SR, (Invitrogen, Thermo Fisher Scientific, Waltham, MA, USA) 5% FBS, 1× (FBS, F7524; Sigma, St. Louis, MO, USA), NEAA-MEM (Sigma Aldrich, St. Louis, MO, USA), 1 μg/mL insulin (Sigma-Aldrich, St. Louis, MO, USA), 0.25 μg/mL hydrocortisone (Sigma-Aldrich, St. Louis, MO, USA), 10 ng/mL EGF (Sigma-Aldrich, St. Louis, MO, USA), and 4 ng/mL bFGF (Sigma-Aldrich, St. Louis, MO, USA) in a humidified atmosphere containing 5% CO_2_ at 37 °C.

### 2.5. Patient Collection of Breast Fibroblast Cells

Primary fibroblast cells of breast were isolated from healthy patients who have undergone mammoplasty reduction. Human mammary cells were isolated from tissues obtained from informed, healthy patients (less than 24 years old), collected with the approval of the Clinical Ethics Committee of the Ministry of Health of Athens, Greece (CEC no. 01072016).

### 2.6. UV Treatment and Evaluation of Apoptotic Cells with Duramycin

Duramycin assay (D-1002—Molecular Targeting Technologies, Inc. West Chester, PA, USA). A Cy5 dye is attached to amino groups of duramycin. The conjugate contains 1 Cy5 dye molecule per duramycin molecule. Duramycin binds phosphatidylethanolamine (PE) at a 1:1 ratio with high affinity (Kd of 4–6 nM) and exclusive specificity. Cells were plated at the concentration of 10,000 cells/cm^2^ in 24 well-plates (Greiner Bio-One, Frickenhausen, Germany) and grown for 24 h. Day after cells were treated with 50 J/m^2^ UVC using UVP HL-2000 HybriLinker incubator and fresh medium was replaced. Untreated cells were used as negative control. After 24 h treatment, treated and untreated cells were washed in 1 × PBS buffer (Sigma-Aldrich, St. Louis, MO, USA) and detached by 0.25% trypsin-0.53 mM EDTA (Invitrogen™ Thermo Fisher Scientific, Waltham, MA, USA) for 3 min at 37 °C. After neutralization with complete medium cells were centrifuged at 0.3 RCF for 5 min and washed once with 1 × Hank’s balanced salt solution (HBSS buffer, Sigma-Aldrich, St. Louis, MO, USA). Supernatant was taken off and cells were re-suspended in 2% FBS-1 × HBSS in 100 µL volume. Untreated and UV-C treated cells were stained in suspension with Duramycin-Cy5 (2 µg/mL) at 37 °C for 20 min, and cell suspension was mixed thoroughly by repeated inversions. After incubation, the reaction was blocked by washing cells twice with 1 mL of 2% FBS-1 × HBS and, centrifuged at 0.2 RCF for 5 min. Cells were resuspended in 200 µL 1 × HBSS and analyzed by flow cytometry (Canto^TM^ II, BD Biosciences, Franklin Lakes, NJ, USA). Detection for Duramycin-Cy5 is through APC. Vertical axis is Side Scatter (SSC).

### 2.7. HER2+ Breast Cancer and Normal Breast Transcriptomic Analysis at the Single Cell Level

#### 2.7.1. Single-Cell RNA Sequencing Data Analysis and Integration of scRNA-Seq Published Datasets

scRNA-seq data from publicly available datasets of adult human breast tissue were downloaded (URL accessed on 30 January 2024) from two separate Chromium 10 × Genomics based studies (Gene Expression Omnibus (GEO) repository: https://www.ncbi.nlm.nih.gov/geo/query/acc.cgi?acc=GSE195665 (accessed 20 July 2025) and data subset GSE235326 for adult healthy breast, and https://www.ncbi.nlm.nih.gov/geo/query/acc.cgi?acc=GSE176078 (accessed 20 July 2025) for human breast cancer patients. Five normal breast samples and 5 HER2+ breast cancer patient samples were selected for this analysis from samples collected from MD Anderson (MDA), UC Irvine (UCI) or Baylor College of Medicine (BCM), USA (see Appendix A summarizing the patient cohort information). HER2+ breast cancer samples were chosen from Caucasian females with no radio or chemo-therapy treatment. In addition to the clinical metadata, adult normal and cancer breast samples were selected based on the procedure by which the tissue source was collected and on the same tissue dissociation protocol and the same single-cell RNA-seq technology (Chromium 10 × Genomics) used. The raw data were imported and processed using the default parameters of the CellRanger pipeline (v.3.1.0, 10 × Genomics), R programming language (v4.4), and Seurat package (v5.1.0). The integrative analysis of the different samples together was performed to allow the data to be combined, and the different cell populations present in the normal and tumor breast tissue to be compared. Osteoarthritis (OA) and rheumatoid arthritis (RA) synovial tissue were obtained by integrating 3 OA https://www.ncbi.nlm.nih.gov/geo/query/acc.cgi?acc=GSE152805 (accessed 20 July 2025) and 4 RA https://www.ncbi.nlm.nih.gov/geo/query/acc.cgi?acc=GSE200815 (accessed 20 July 2025) datasets as previously described [1].

#### 2.7.2. Data Normalization, Integration and Clustering

Normalization and log-transformation of each dataset was with Seurat v3.0.0 and Method36 in R (v3.5.0) as described in our previous study of *RNASET2*, syndecans and *TMEM230* for patients with rheumatoid arthritis [4]. Dimensionality reduction and clustering were performed using default parameters. Integration anchors were calculated with Find Integration Anchors. Datasets were integrated using Integrate Data (Seurat V.5.2.0). Downstream analysis was performed on the integrated assay, scaled using Scale Data (Seurat V5.2.0) and reduced in dimensionality using PCA and UMAP (Run PCA, RunUMAP). Clusters were identified using the Louvain algorithm Find Neighbors. Find Clusters (Seurat V5.2.0) at a resolution of 0.5 following construction of a shared nearest neighbor graph [19,20].

#### 2.7.3. Visualization

UMAP projections were formed to see cells by sample, cluster identity, and disease condition using DimPlot. Cluster identities were manually annotated based on known markers and renamed using RenameIdents. Violin plots and additional feature plots (VlnPlot, FeaturePlot, Seurat V5.2.0) were created to visualize gene expression patterns across clusters.

#### 2.7.4. Differential Expression and Cell Type Composition

Differential expression analysis within individual clusters was by using Seurat Find Markers function and the Wilcoxauc method (pRESTO package v5.3.0) for disease and normal samples. Markers were selected by minimum expression percentage and adjusted *p*-value thresholds. Bar plots were generated to display relative proportions, number of cells per sample and number of cells per cluster (ggplot2).

#### 2.7.5. Gene Set Enrichment Analysis (GSEA)

Gene Set Enrichment Analysis was performed on ranked gene lists (by AUC) for each cluster using the fgsea package. Gene sets were obtained from the MSigDB database via *msigdbr,* spanning collections H, C1–C8. Enrichment scores and leading-edge genes were calculated and classified as upregulated or downregulated based on AUC thresholds.

## 3. Results

### 3.1. Comparative IRF1 mRNA Expression in RA and OA Cell Clusters of Synovial Tissue from Patients and Identification of Stress and Cell Defense Response Genes and Pathways

We previously demonstrated that *TMEM230* and *RNASET2* were downregulated in *CXCL12* expressing synovial fibroblast cells of rheumatoid arthritis (RA) compared to osteoarthritis (OA) patients [1]. *CXCL12* positive cells contribute to tissue remodeling in RA by *TMEM230* dependent trafficking and secretion of membrane bound vesicles containing *RNASET2* and syndecans. In the present study we further investigated whether other pathways were downregulated with *TMEM230* modulation in *CXCL12* expressing synovial fibroblast cells of RA patients (Table 1). Pathways in response to reactive oxygen species, endoplasmic reticulum unfolded proteins, xenobiotic metabolism, ultraviolet (UV) radiation, and hypoxia were identified, supporting that *TMEM230* has diverse roles in stress and defense response and that *TMEM230* may contribute to several human pathological conditions (Table 1). As we previously identified that many of these pathways in gliomas of patients were associated with aberrant expression of *TMEM230* and *RNASET2*, in our current study we investigated whether these pathways would also be identified in fibroblast clusters of *HER2*+ breast cancer patients.

We and other groups have previously identified that the interferon induced transmembrane protein 3 (*IFITM3*) and the interferon (IFN) pathways are co-regulated with the tumor protein *TP53* pathway components, such as p21 (*CDKN1A*) [12]. The identification of *TP53* pathway in *CXCL12* fibroblast cells in RA (Table 1) supported our previous results showing that interferons (IFNs) modulate glycosylation and trafficking of major histocompatibility complex (MHC) antigens in RA [2]. 

By screening for differentially expressed IRFs, *IRF1*, a trans activator of *TP53* was identified differentially expressed in RA in respect to OA patients (Appendix A and Figure 1). While the fibroblast cell clusters (SF_1, *CXCL12*_SF, and *PRG4*_SF) displayed an extreme range of expression of *IRF1* in uniform manifold approximation and projection (UMAP) analysis, (blue (high) and grey (low), Appendix A) in outlier cells (Figure 1), *IRF1* was most downregulated (approximately 4 times, *p*-value ≤ 0.05) in *CXCL12* fibroblasts (indicated by a red asterisk).

To support the hypothesis that *IRF1* may have a role in fibroblast cells from human breast tumors, in vitro cell assays were performed in patient-derived fibroblast cells by constitutive interference of the wild type *IRF1* gene activity.

### 3.2. Generation of Lentiviral Construct Expressing IRF1 Intron 4 Retaining (IRF1+I4) Isoform and Cell Culture Assays of Fibroblast Cells from Patients with No History of Tumor

Deletions or rearrangements of the IRF-1 tumor suppressor gene and exon skipping of the *IRF*-*1* full-length wild type (*IRF*-*1* WT) transcript have been described in patients with myelodysplastic syndromes (MDS) and leukemia [21,22,23,24,25,26,27,28,29,30,31]. An isoform of *IRF1* was identified, designated *IRF1*+*Intron4* (*IRF1*+*I4*, GenBank KC209828.1 and Appendix A) having dominant negative activity on the wild-type IRF1 protein. As the UV induced cell damage response pathway (https://www.gsea-msigdb.org/gsea/msigdb/cards/HALLMARK_UV_RESPONSE_UP) (accessed 20 July 2025) was identified in *CXCL12* cells (Table 1) to be regulated by *TP53* and IFNs, expression and tissue culture assays were performed using breast fibroblast cells isolated from patients with no history of cancer. In these cells, normal *IRF1* activity was inhibited by constitutive expression of the *IRF1*+*I4* isoform using a lentiviral system. As expected, inactivation of IRF1 promoted downregulation of *TP53* expression (Appendix A). In agreement, as *IRF1* transactivates *TP53* and the primary target of the *TP53* is p21, a cyclin dependent kinase (CDK) inhibitor, the expression of p21 was investigated and it was also found downregulated (Appendix A). Both genes induce cell cycle arrest, thereby allowing DNA repair mechanisms to be activated or alternatively, they promote programmed cell death (apoptosis) if DNA repair cannot be accomplished [32,33]. 

The roles of *TP53* and *IRF1* are therefore complex and appear contradictory as we have previously demonstrated that p21 can promote cell cycle arrest of cancer stem cells (CSCs) [34]. Induction of a cell quiescent state is thought to allow CSCs to survive the DNA damaging effects of anti-cancer drugs [12,22]. However, p21 inhibition of cyclin dependent kinase 2 (*CDK2*) expression can also promote apoptosis of CSCs. In agreement with the observation that p21 inhibits CDK2, *CDK2* was found inversely expressed in respect to p21 in *IRF1*+*I4* transduced cells (Appendix A) [35]. The upregulation of *CDK2* suggests that *IRF1* has a role in apoptosis in human breast fibroblast cells. However, upregulation of *CDK2* is associated with both inhibiting and promoting apoptosis. 

We therefore examined the expression level of *BIRC5* (survivin) in *IRF1*+*I4* transduced cells. *BIRC5* is a member of the inhibitor of apoptosis family of genes. Like RNASET2, BIRC5 is reported to be expressed in a *TP53*/p21 cell cycle dependent manner and to repress caspase activation and therefore apoptosis [36,37]. In agreement that *BIRC5* is expressed highly in most human tumors and completely absent in terminally differentiated cells, *BIRC5* was found upregulated in *IRF1*+*I4* transduced breast derived fibroblast, suggesting that loss of IRF1 activity inhibits apoptosis (Appendix A). As the expression data does not provide evidence for a conclusive role of *IRF1* in breast cancer, to determine whether downregulation of *IRF1* was unambiguously associated with change in cell number and apoptosis, *IRF1*+*I4* transduced fibroblast cells from patients with no tumor history were investigated in cell culture assays.

The cell culture assays showed decrease in cell number in cells expressing *IRF1*+*I4* compared to control (Figure 2 and Appendix A).

To ascertain the activity of *IRF1* in apoptosis, *IRF1*+*4I* transduced fibroblast cells derived from patients without tumors were then treated with UV radiation to induce cell stress. Apoptotic and necrotic cells were quantified using fluorescent activated flow cytometry with duramycin that binds phosphatidylethanolamine present in the cell membrane. Differences in intensity of the fluorophore associated with duramycin identified which cells were undergoing apoptosis or necrosis. Breast fibroblast cells transduced with *IRF1*+*I4*, when cultured after treatment with UV radiation, were observed to have increased apoptotic activity (*p* < 0.05) compared to control *GFP* cells (Appendix A). In a representative analysis, 1.6% of control cells expressing wildtype IRF1 were associated with duramycin expression (population P3) compared to 3.8% of IRF1+I4 cells.

In conclusion, our data showed that loss of IRF1 expression with transduction of IRF1+I4 was associated with decrease of *TP53* and p21, increase of *CDK2* and BIRC5 expressions, and increased apoptosis (Appendix A). 

Paradoxically, while the expression analysis showed inhibition of apoptosis, the in vitro cell culture assays showed increase of apoptosis with loss of *IRF1* in breast fibroblast cells (Figure 2). 

Our previous studies supported that in addition to *IRF1*, *TMEM230*, *RNASET2*, and *SDC2* have tumor promoting and inhibiting functions. To explain this, we therefor hypothesized that the oncogenic activity of genes with pleiotropic functions is determined by which cell types these genes are expressed in. Additionally, different levels of expression may promote or inhibit oncogenic activity. Therefore, to exactly determine the tole of genes with pleiotropic functions it is necessary to assay all cell types in a tumor mass. The technology of single cell sequencing provides a powerful tool to correlate the expression level of a gene with different molecular pathways in diverse cell types in human tumor [19,20]. 

For instance, the IFNγ pathway regulated by IRF1 is reported to have pleiotropic functions in tumors by suppressing and inducing apoptosis [38,39,40,41,42]. We propose that this apparent contradiction can be due to previous research having mostly been performed using clonal cell lines of a specific cell type which do not recapitulate the interactions of all the different cell types in a tumor tissue in which IFNs are expressed. Our single cell transcript analysis of synovial tissue of RA patient for instance showed definitively that different types of fibroblast cells have significantly different levels of expression of IRF1 and therefore some fibroblast cells may promote or inhibit apoptosis (Figure 1).

### 3.3. IRF1 Expression in Human HER+ Breast Tumor Tissue by Single Cell Transcriptomic Analysis

To ascertain which cell types may be associated with expression and modulation of *IRF1* expression in breast tumor, *IRF1* expression was analyzed by single cell RNA sequencing in all cell types derived from *HER2*+ (*ERBB2*) tumor and non-malignant breast tissues from patients, used as control (Figure 3).

In normal breast tissue, IRF1 was found more expressed in lymphoid cells (Lymph), in the UNC2 uncharacterized cell cluster, in different types of fibroblasts (FB2, FB1), and in the smooth muscle (SM) cells (Figure 4 and Figure 5). *IRF1* was downregulated in *HER2*+ tumor FB2 fibroblast and smooth muscle cells and not detected in a cluster of relatively rare epithelial cells indicated as EPI2 (Figure 6). In contrast, *IRF1* was found upregulated in basal and in the uncharacterized cell cluster UNC2 in *HER2*+ tumors (Figure 6). Additionally, IRF1 was expressed in the uncharacterized cell clusters UNC1 and UNC4 in *HER2*+ tumors but not in normal breast tissue (Figure 6). 

These expression patterns supported our hypothesis and suggested that the potential antagonistic oncogenic activities of *IRF1* may be due to *IRF1* having different levels of expression in different cell types of the same tumor.

In our previous study IRF1 was identified as co-regulated with or regulated by *TMEM230* in fibroblast cells of RA patients [1,2]. As *TMEM230* is necessary for glycosylation and secretion of proteoglycans and glycoproteins with roles in cell stress and defense response, we analyzed expression of *RNASET2* and syndecans (SDCs) in *HER2*+ tumors.

### 3.4. Expression Profile and Cluster Localization of the TMEM230 Glycosylated SDC2 and RNASET2 Genes in HER2+ Tumors

We previously demonstrated that syndecan transmembrane proteins were regulated by *TMEM230* in autoimmunity and GBM tumors [43,44]. Syndecan 2 (*SDC2*), like *IRF1* participates in fibroblast cell proliferation and cell growth in stress response. Both *SDC2* and *IRF1* were found highly expressed in mast cells and FB2 fibroblast cells in normal breast tissue (Figure 7 and Figure 8).

*IRF1* and *SDC2* were downregulated in FB2 and not detected in epithelial EPI2 cells in *HER2*+ tumors (Figure 6 and Figure 9). In contrast to *IRF1*, *SDC2* was upregulated in smooth muscle (SM) cells in HER2+ tumors (Figure 9). *SDC2* was not found differentially expressed in LUM_SEC and LUM_HR epithelial luminal cells (Figure 9). This suggested that if *SDC2* has a role in cancer, it is not in the luminal cells, but in mast cells and in FB2 fibroblast cells. In *CXCL12* synovial fibroblast cells from RA patients, *SDC2* was found to have a role in extracellular matrix remodeling, suggesting that IRF1 may have a similar role in cancer (Figure 1). Chemokines with C-X-C motifs such as *CXCL12* are chemoattractants for immune system cells and are normally localized to sites of injury and infections and play a role in inflammatory processes [45].

We previously determined *TMEM230* had a role in the regulation of antigen processing, transport, and antigen presentation [1,2]. Antigen processing and presentation are dependent on *TMEM230* regulating the RNA digesting enzyme RNASET2, a component of lysosomes. Like *SDC2*, *RNASET2* contributes tissue remodeling in *CXCL12* fibroblasts [45]. Additionally, *RNASET2* like *IRF1* is closely linked with immune responses in viral infections and cancer [2,46,47,48]. We therefore investigated the possible role of *RNASET2* with IRF1 in breast cancer (Figure 10 and Figure 11).

In normal breast, highest expression of *RNASET2* was found predominantly in cells associated with immune functions such as plasmacytoid, macrophage (MP), dendritic cells (DC), B, T and plasma cells (Figure 10). Additionally, *RNASET2* was expressed in the luminal hormone responding (LUM_HR) cluster, in the epithelial (EPI2) cluster and in the luminal secreting cell cluster (LUM_SEC). These cells function as tissue barrier cells in lumen structure protection in infection. In agreement with its role in immune cell function, *RNASET2* was downregulated or absent in EPI2 epithelial cell cluster in *HER2*+ breast tumors, as seen for the expression pattern of *IRF1* (Figure 6 and Figure 11).

### 3.5. Identification of Molecular Pathways Differentially Regulated in HER2+ Tumor and Control Cell Clusters

To better understand the molecular pathways involved in cell stress and defense response that may contribute to tumor development, we performed gene pathway analysis (Table 2, Table 3, Table 4, Table 5, Table 6 and Table 7). Many of the pathways regulated by the *TMEM230*/*RNASET2*/*SDC2*/*IRF1* in *CXCL12* fibroblast cells from RA patients including oxidative phosphorylation, MYC target genes, DNA repair, *TP53*, and metabolism (adipogenesis, glycolysis, and fatty acid synthesis and breakdown) (Table 2 and Table 3) were also identified in the FB2 fibroblast cell cluster from *HER2*+ breast cancer patients. These pathways are active or influenced by transmembrane protein, TMEM230 in the endoplasmic reticulum (ER). For instance, IFN alpha and gamma regulate ER dependent function in antigen presentation, suggesting that the *TMEM230*/*RNASET2*/*SDC2*/*IRF1* axis is mis regulated both in autoimmunity and cancer.

Pathways identified in *CXCL12* and FB2 fibroblast cell clusters from RA and in the B2 cell cluster from *HER2*+ breast cancer patients support that *TMEM230*/*RNASET2*/*SDC2*/*IRF1* axis have pleiotropic functions (Table 3). The pathways uncovered suggest that the role of *IRF1* in the FB2 fibroblast cell cluster can be summarized into regulation of metabolism (glycolysis, adipogenesis, mitochondrial energy production with oxidative phosphorylation, and fatty acid metabolism), tissue remodeling (EMT, angiogenesis, coagulation, and apical junction) and cell and immunity associated stress responses (*TP53*, hypoxia, *MYC*, IFNs, and allograft rejection) (Table 3).

Pathways including epithelial to mesenchyme transition (EMT), apical junction, angiogenesis, and coagulation regulation were modulated with *IRF1* downregulation in the *HER2*+ breast cancer FB2 fibroblast cell cluster (Table 3). 

The gene pathway analysis also identified previously unknown pathways in IRF1 signaling (Table 2, Table 3, Table 4, Table 5, Table 6 and Table 7). For instance, the IFN-gamma (Table 2, Table 4, and Table 7 in FB2, basal and immune system associated cells) and the tumor necrosis factor, TNF-alpha signaling pathways (Table 4 and Table 7 in basal and immune system associated cells) that have functions in modulating cell growth were uncovered. The *IRF1* pleiotropic activity is indicated by the increased expression of *IRF1* in *HER2*+ breast tumor basal cells (Table 4 and Table 5) that in contrast to FB2 cells, were associated with the *IL6*-*JAK*-*STAT3* signaling pathway.

In contrast to FB2 fibroblasts, *IRF1* was not expressed in the uncharacterized cell clusters, UNC1 and UNC4 in normal breast tissue, but it was upregulated in both clusters in tumor tissue (Table 6). In the other cell types, *IRF1* was expressed in epithelial EPI2 cells in normal breast but was not detected in *HER2*+ tumors, suggesting that the loss of these cells may promote tumor formation. 

Collectively the pathways identified makes clear that single cell RNA sequencing is a powerful platform for uncovering the complexity of cell and gene interactions in tumor containing diverse cell types. Previous studies using clonally established tumor cell lines could not provide an understanding of the behavior of different cell type interactions in tumors. Similarly, sequencing of whole tumor tissue may not uncover transcriptome profiles induced by the interactions of any specific cell type within a tumor as it would be not clear which genes and pathways are associated with which cell type.

While *RNASET2* like *IRF1* is closely linked with immune responses in viral infections and like *IRF1* is thought to have a role in cancer, *RNASET2* was not identified coregulated with *IRF1* in FB2 cells. We therefore investigated the possible pathways regulated by *RNASET2* in other cell types found in *HER2*+ breast tumors (Table 7). Largest difference in the expression of *RNASET2* between control and *HER2*+ breast tumors was observed in the uncharacterized epithelial EPI2 cell clusters and in clusters associated with immune functions such as B, natural killer (NK) and plasma cells (Figure 11). Like *IRF1*, *RNASET2* was downregulated in EPI2 (Figure 6) suggesting that these uncharacterized rare cells are linked with *RNASET2* activity in oncogenesis. *RNASET2* was differentially expressed in the UNC4 and EPI2 cell clusters but the number of cells was too few to identify in these clusters molecular pathways differentially regulated between *HER2*+ tumor and non-malignant breast. Pathways associated with differential expression of *RNASET2* in cells associated with immune functions such as B, natural killer (NK) and plasma cells displayed many pathways including the interferon response, TNFA (TNF-α) and apoptosis pathways (Table 7). Additionally, *RNASET2* was also identified having a role in the regulation of cell stress and defense response, such as the mammalian target of rapamycin complex (*MTORC*) in B cells. MTORC regulates metabolism as nutrients, energy, redox sensor primarily through lysosome proteins such as RNASET2 [2].

## 4. Conclusions

TMEM230, a transmembrane protein is a master regulator of the endoplasmic reticulum (ER) [49]. As the ER has developed diverse functions during eukaryotic evolution, not surprisingly *TMEM230* has pleiotropic functions including the regulation the initial steps of glycosylation in glycoprotein and proteoglycan formation, and their intracellular trafficking and secretion. Glycosylation and glycoconjugate trafficking are essential in immunity and cell stress and defense responses of all tissue cells [1,2]. Stress responses associated with the ER are well represented in the pathways identified in RA and *HER2*+ breast tumors, as we have shown and include *MTORC*, unfolded protein response, *TP53*, IFN and interleukin regulation (Table 1, Table 2, Table 3, Table 4, Table 5, Table 6 and Table 7). Without the immense molecular diversity that glycans generate on molecules, host cell self-proteins and pathogen recognition would not be possible [2]. Loss of proper regulation of the diverse ER functions by TMEM230 is therefore expected to contribute to diverse human pathologies. For instance, TMEM230 regulated glycoproteins and proteoglycans, RNASET2 and SDC2 regulate cell-to-cell and cell-to-extracellular matrix interactions and remodeling in wound healing, inflammation and immune system activities. *TMEM230*, *RNASET2* and IFNs regulate antigen processing, trafficking and presentation in autoimmunity. Sequencing analysis of fibroblast cells from patient with tumors uncovered interferon alpha and gamma pathways with downregulation of *IRF1* (Table 2). IFNs promote antigen presentation by increasing expression of major histocompatibility complex (MHC) couple antigens. We observed that like *IRF1*, *SDC2* was downregulated in FB2 and downregulated in epithelial EPI2 cells in *HER2*+ tumors (Figure 9). *SDC2* was not found differentially expressed in LUM_SEC and LUM_HR epithelial luminal cells (Figure 9) suggesting that the role of *SDC2* in cancer is in FB2 fibroblast cells. In contrast to *SDC2*, *RNASET2* appears to be linked with *IRF1* not in FB2 cells but immune system associated cells (see Figure 6). The unexpected finding was uncovered by single cell transcript analysis, demonstrating the power of this unique platform.

Much of the early publications in identifying the role of IFNs was by using clonally derived specific cell types or global transcript sequencing of the entire tumor tissue, platforms that could not characterize collectively the expression levels of *IRF1* in each cell type. Our study is unique as we characterized *IRF1* expression in all cell types in breast tissue of *HER2*+ patients in contrast to previous studies that have utilized breast bulk patient samples in toto or tumor epithelial cell lines. Established cell lines often do not recapitulate the behavior of native cells in normal and tumor tissue. Previous literature suggested that the apparent contradictory, pleiotropic roles of *IRF1* are likely due to IRF1 having a vast number of DNA binding site target genes. For instance, a previous study using cancer cell lines have uncovered that cell treatment with IFN-gamma was associated with more that 170,000 DNA binding events associated with *IRF1* (see excellent review Perevalova et al. 2024 [8]). This rationale was used to explain why IFNs therefore have both inhibitory and promoting functions. Here we show that the ultimate oncogenic outcome for a tumor mass may be due to the collective interactions of all cell types expressing a specific gene with pleiotropic functions. Here we have uncovered that in addition to IFNs being associated with vast number of DNA binding sites, the mis regulation of protein folding performed in the ER by the *E2F* pathway may contribute to cancer by secreting aberrantly folded extracellular factors such as *SDC2*, that have roles in immune and tissue interactions. Additional pathways uncovered *(*Table 3) indicated that different pathways are activated with IRF1 depending on the cell type. 

Our study provides insight into how *IRF1* differentially expressed in different cell types of the same tumor may in part contribute to the pleiotropic functions observed previously in other studies, that is IRF1 displaying both antitumor and pro-tumor induction properties. In conclusion, our results may help explain why *IRF1* may induce or inhibit apoptosis in different cell types in tumors and may help uncover novel IFN based therapies.

## Figures and Tables

**Figure 1 cells-14-01246-f001:**
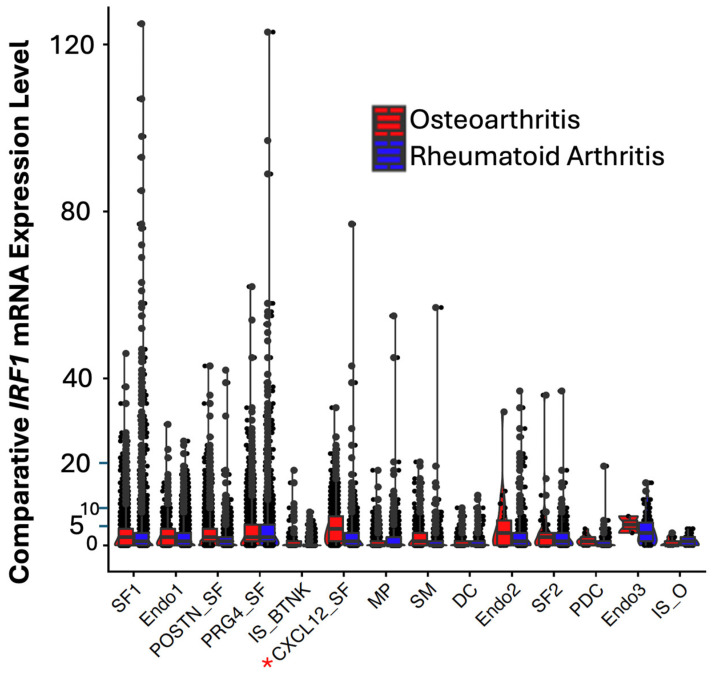
Comparative IRF1 mRNA expression in RA and OA cell clusters from synovial tissue. IRF1 mRNA was downregulated (4 times, *p*-value ≤ 0.05) in CXCL12 expressing synovial fibroblast cells of RA (see Appendix A). Asterisk represents the fibroblast cell cluster in which *IRF1* was significantly modulated.

**Figure 2 cells-14-01246-f002:**
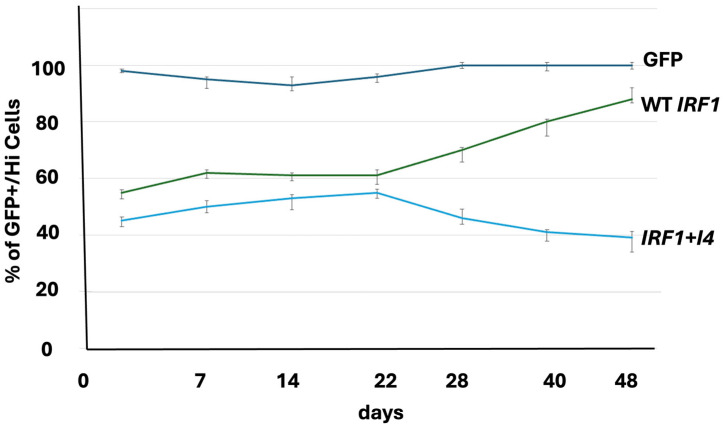
Cells expressing *GFP* alone (*GFP* control), WT-*IRF1*-*GFP* (WT *IRF1* control) or *IRF1*+*I4*-*GFP* (*IRF1*+*I4*) were counted with fluorescent activated flow cytometry in multiweek cell culture assays. Decrease in cell number was correlated with *IRF1*+*I4* expression in fibroblast cells of patients without tumors.

**Figure 3 cells-14-01246-f003:**
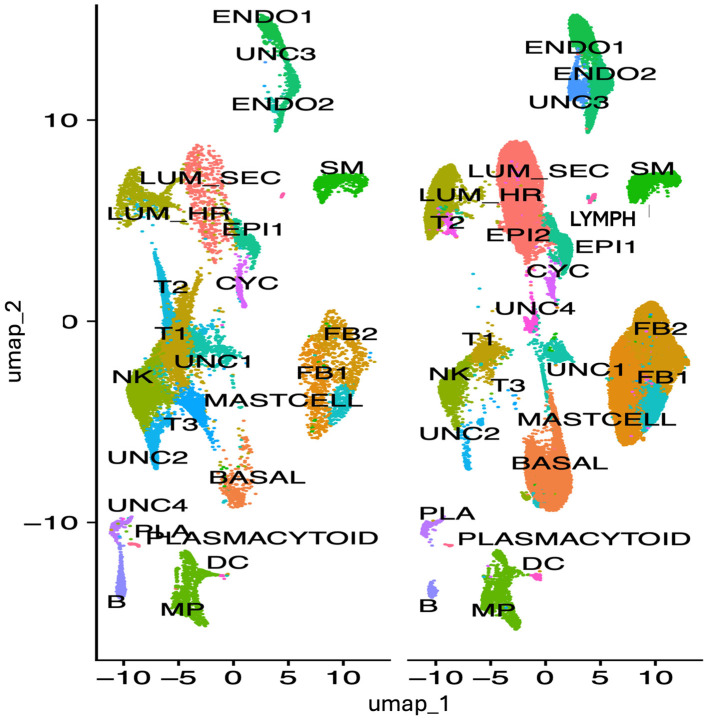
HER2+ breast tumor and control breast tissue transcriptomic map clustered by cell type. High-resolution visualization of tissue composition obtained by Uniform Manifold Approximation and Projection (UMAP) plot of tissue cell type after integrating the expression profiles of five normal breast samples and 5 HER2+ samples. scRNA-seq data from publicly available datasets of adult human breast tissue were downloaded from Gene Expression Omnibus (GEO) repository: https://www.ncbi.nlm.nih.gov/geo/query/acc.cgi?acc=GSE195665 (accessed 20 July 2025) for adult normal healthy breast and https://www.ncbi.nlm.nih.gov/geo/query/acc.cgi?acc=GSE176078 (accessed 20 July 2025) for human breast cancer patients. Different clusters were indicated by different colors according to the expression of representative specific markers reported [19,20].

**Figure 4 cells-14-01246-f004:**
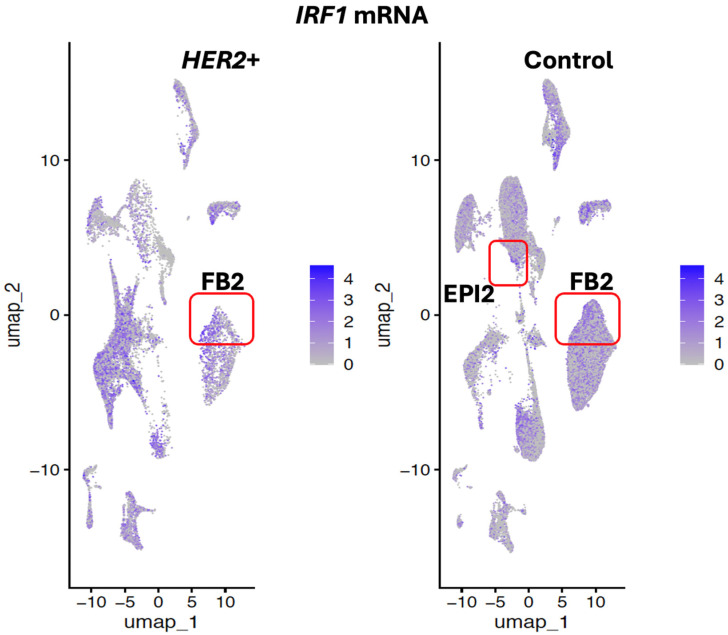
Expression profile and localization of IRF1 in cell clusters of *HER2*+ tumor and control breast tissue. Cell clusters were defined by markers from Figure 3. Colors indicate the levels of expression from 1 (gray) to 3 (blue), low to high. Fibroblast cluster FB2 and uncharacterized epithelial cluster EPI2 are indicated with red boxes.

**Figure 5 cells-14-01246-f005:**
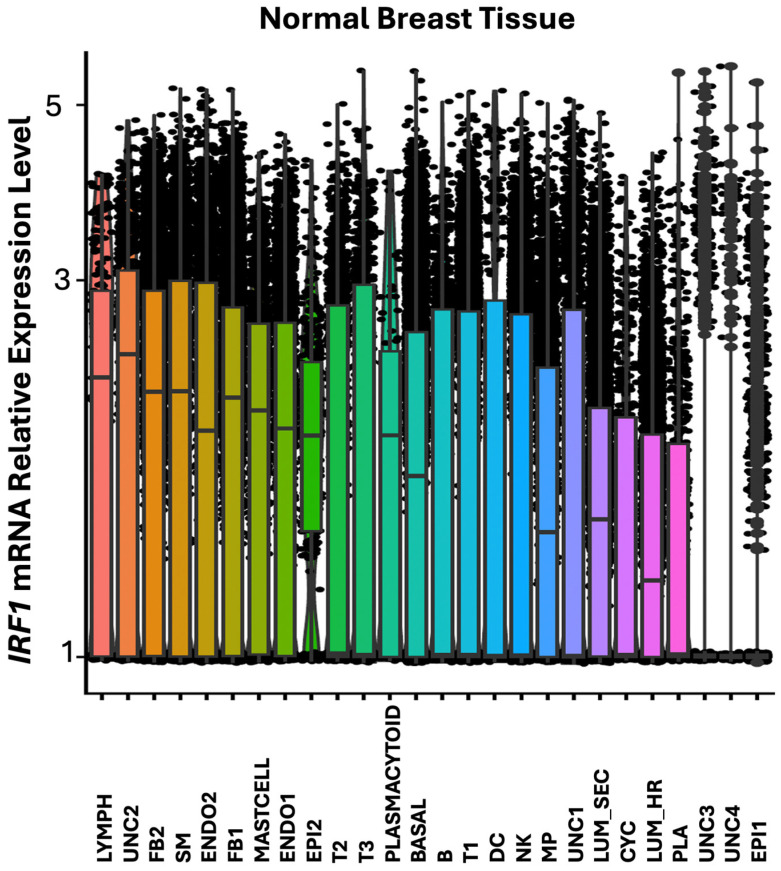
Box plot graphical visualization of the expression profile of the *IRF1* in normal breast tissue. The y axis represents the expression level. The x axis represents the 26 cell clusters identified in normal breast tissue. Colors indicate different clusters of cell types.

**Figure 6 cells-14-01246-f006:**
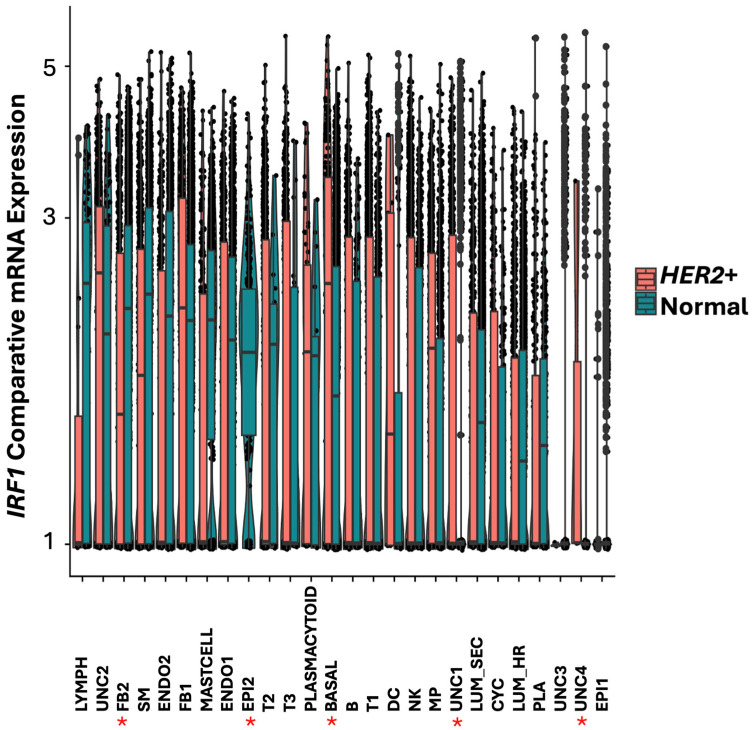
Box plot graphical visualization of the expression profile of IRF1 in cell clusters of *HER2*+ breast tumor and control breast tissue. The y axis represents expression levels. *p* value < 0.05 was calculated by Kolmogorov Smirnov (K-S) test. Cell clusters in which *IRF1* were significantly modulated in *HER2*+ and normal tissue are shown with asterisks.

**Figure 7 cells-14-01246-f007:**
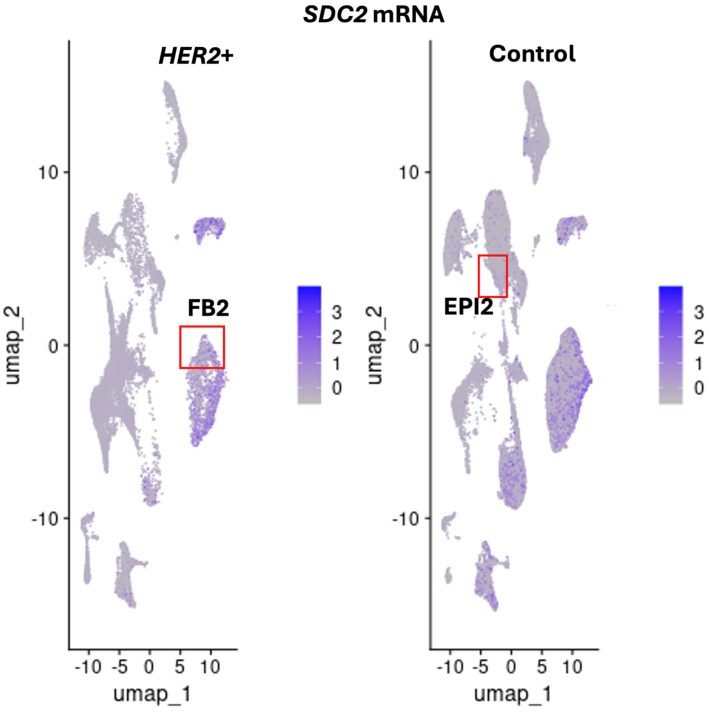
Expression profile and localization of *SDC2* in *HER2*+ tumor or control cell clusters, with cell clusters defined by markers from Figure 3. Colors indicate the levels of expression from 1 (gray) to 3 (blue), low to high. Fibroblast cluster FB2 and uncharacterized epithelial cluster EPI2 are indicated with red boxes.

**Figure 8 cells-14-01246-f008:**
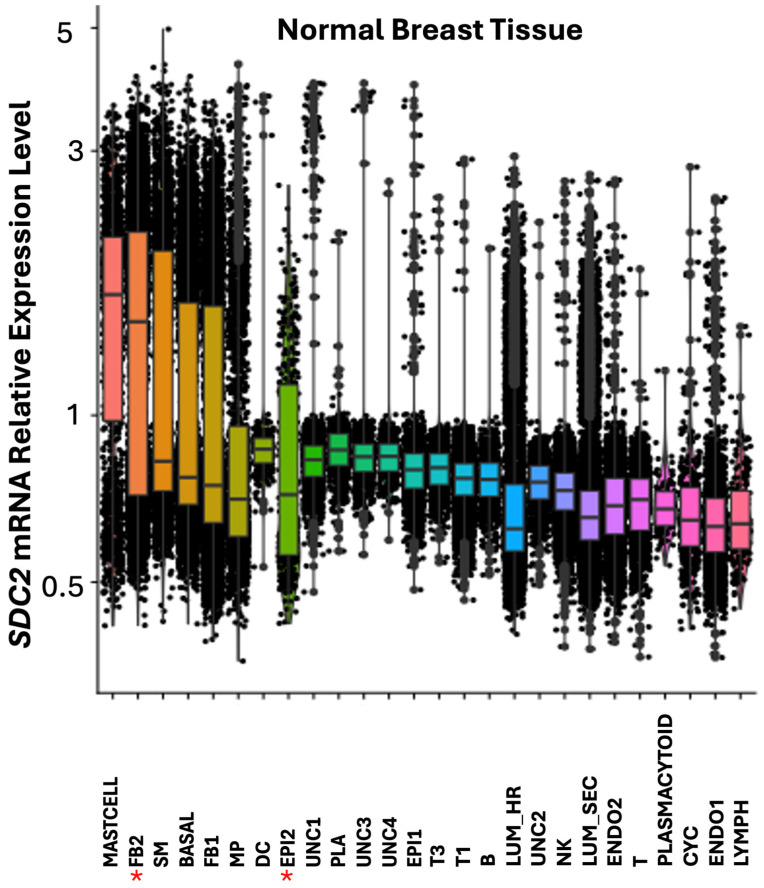
Box plot graphical visualization of the expression profile of the SDC2 in normal breast tissue. The y axis represents the expression level. Asterisks represent cell clusters in which IRF1 was downregulated or absent in *HER2*+ tumors (Figure 6). Colors indicate different clusters of cell types.

**Figure 9 cells-14-01246-f009:**
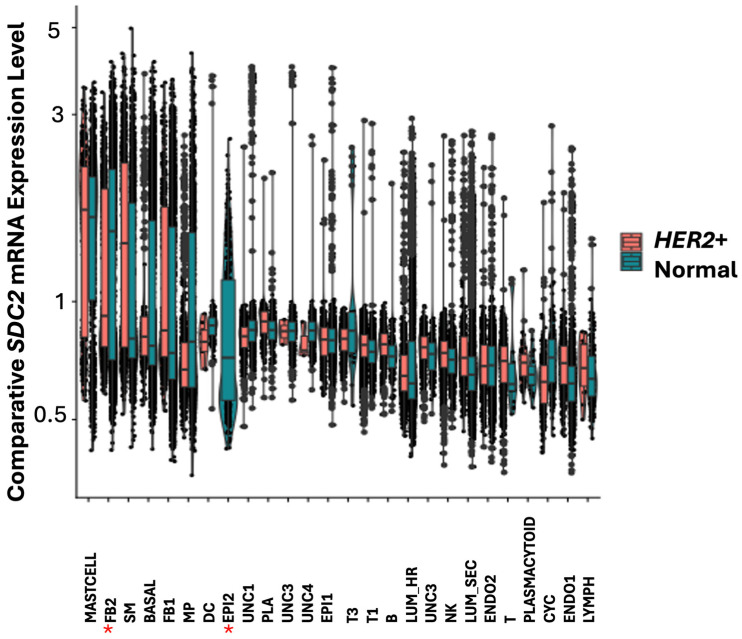
Box plot graphical visualization of the expression profile of SDC2 in cell clusters of HER2+ and control breast tissue. The y axis represents the expression level. *p* value < 0.05 was calculated by Kolmogorov Smirnov (K-S) test. Asterisks represent cell clusters in which IRF1 was downregulated or absent in HER2+ tumors (Figure 6).

**Figure 10 cells-14-01246-f010:**
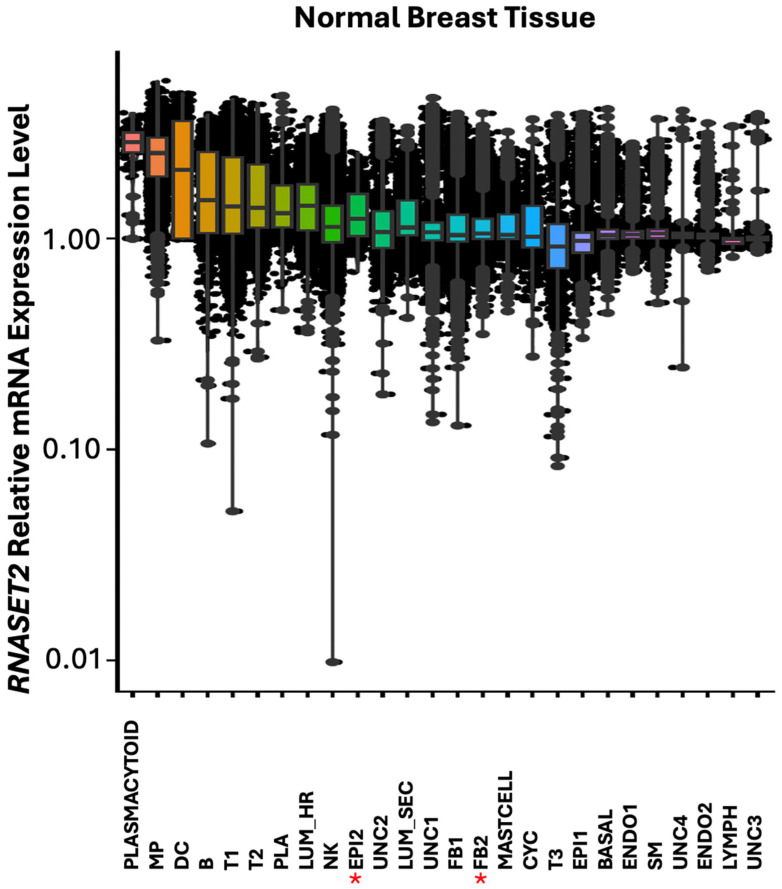
Box plot graphical visualization of the expression profile of theRNASET2 in normal breast tissue. The y axis represents the expression level. *p* value < 0.05 was calculated by Kolmogorov Smirnov (K-S) test. Asterisks represent cell clusters in which IRF1 was downregulated or absent in HER2+ tumors. Colors indicate different clusters of cell types.

**Figure 11 cells-14-01246-f011:**
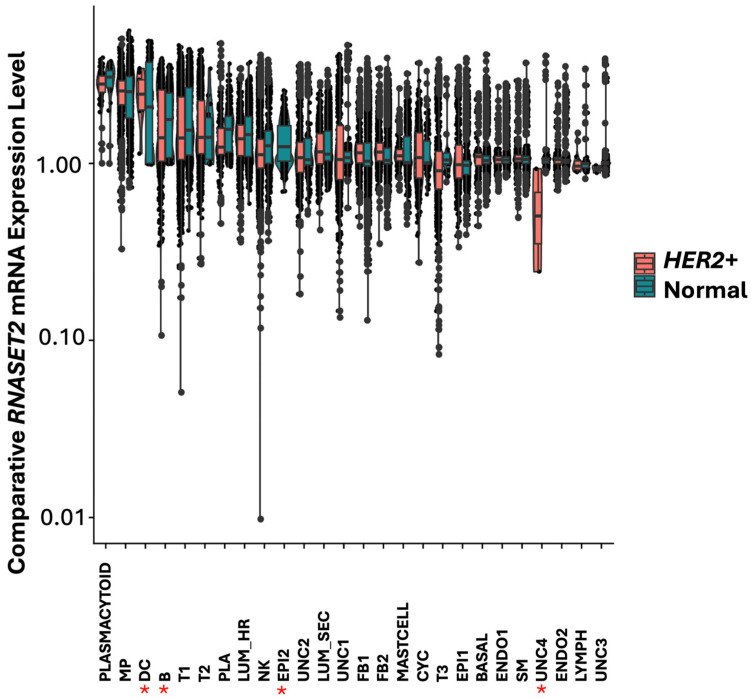
Box plot graphical visualization of the expression profile of *RNASET2* in cell clusters of *HER2*+ and control breast tissue. The y axis represents the expression level. *p* value < 0.05 was calculated by Kolmogorov Smirnov (K-S) test. Asterisks represent cell clusters in which IRF1 was significantly modulated in *HER2*+ tumors compared to control tissue. *RNASET2* was downregulated or absent in EPI2 epithelial cell cluster in tumors as seen for the expression pattern of IRF1 (Figure 6) suggesting that these uncharacterized rare cells link *RNASET2* with *IRF1* activities in *HER2*+.

**Table 1 cells-14-01246-t001:** Gene Set Enrichment Analysis (GSEA) identified molecular pathways differentially regulated with downregulation of *TMEM230* expression in *CXCL12* synovial fibroblast cells of RA compared to OA patients.

*CXCL12* Synovial Fibroblast Cells Pathways in Autoimmunity	pval	padj
HALLMARK_OXIDATIVE_PHOSPHORYLATION	0.001040583	0.00794155
HALLMARK_MYC_TARGET	0.00104612	0.00794155
HALLMARK_DNA_REPAIR	0.001108647	0.00794155
HALLMARK_REACTIVE_OXIGEN_SPECIES_PATHWAY	0.001270648	0.00794155
HALLMARK_ADIPOGENESIS	0.001072961	0.00794155
HALLMARK_FATTY ACID METABOLISM	0.00116195	0.009238729
HALLMARK_UNFOLDED_PROTEIN_RESPONSE	0.001146789	0.00794155
HALLMARK_XENOBIOTIC_METABOLISM	0.002107802	0.009238729
HALLMARK_MTORC1_SIGNALING	0.002162162	0.009238729
HALLMARK_E2F_TARGETS	0.002202643	0.009238729
HALLMARK_UV RESPONSE_UP	0.002217295	0.009238729
HALLMARK_COAGULATION	0.001183432	0.00794155
HALLMARK_GLYCOLYSIS	0.004338395	0.015494267
HALLMARK_HYPOXIA	0.00867679	0.016287
HALLMARK_MYOGENESIS	0.01818188	0.037037037
HALLMARK_MYC_TARGETS_V2	0.016290727	0.037024379
HALLMARK_P53_PATHWAY	0.023809524	0.044091711
HALLMARK_PEROXISOME	0.025	0.044642857

**Table 2 cells-14-01246-t002:** Selected biological pathways differentially regulated in the *HER2*+ and control FB2 fibroblast cells.

FB2 Fibroblast Pathways in *HER2*+ Breast Tissue	pval	padj
HALLMARK_OXIDATIVE_PHOSPHORYLATION	0.00103	0.007135
HALLMARK_EPITHELIAL_MESENCHYMAL_TRANSITION	0.001031	0.007135
HALLMARK_INTERFERON_ALPHA_RESPONSE	0.001142	0.007135
HALLMARK_DNA_REPAIR	0.001063	0.007135
HALLMARK_COAGULATION	0.01142	0.007135
HALLMARK_GLYCOLYSIS	0.001062	0.007135
HALLMARK_INTERFERON_GAMMA_RESPONSE	0.001044	0.007135
HALLMARK_APICAL JUNCTION	0.001089	0.007135
HALLMARK_ADIPOGENESIS	0.002079	0.010893
HALLMARK_FATTY ACID METABOLISM	0.002179	0.010893
HALLMARK_ANGIOGENESIS	0.011952	0.037707
HALLMARK_P53_PATHWAY	0.003128	0.013034
HALLMARK_MYC_TARGETS_V1	0.003093	0.013034
HALLMARK_HYPOXIA	0.019958	0.049895
HALLMARK_ALLOGRAFT_REJECTION	0.012586	0.037707

**Table 3 cells-14-01246-t003:** Biological pathways in common or differentially regulated in *CXCL12* synovial fibroblasts of RA and FB2 fibroblast cells of *HER2*+ breast tumor patients. Bold indicates pathways in common.

*CXCL12* Synovial Fibroblast Pathways	pval	padj	Fb2 Fibroblast Pathways in *Her2*+ Breast Tissue	pval	padj
**HALLMARK_OXIDATIVE_PHOSPHORYLATION**	**0.001040583**	**0.00794155**	**HALLMARK_OXIDATIVE_PHOSPHORYLATION**	**0.00103**	**0.007135**
HALLMARK_MYC_TARGET	0.00104612	0.00794155	HALLMARK_EPITHELIAL_MESENCHYMAL_TRANSITION	0.001031	0.007135
**HALLMARK_DNA_REPAIR**	0.001108647	0.00794155	HALLMARK_INTERFERON_ALPHA_RESPONSE	0.001142	0.007135
HALLMARK_REACTIVE_OXIGEN_SPECIES_PATHWAY	0.001270648	0.00794155	**HALLMARK_DNA_REPAIR**	0.001063	0.007135
**HALLMARK_ADIPOGENESIS**	**0.001072961**	**0.00794155**	**HALLMARK_COAGULATION**	**0.01142**	**0.007135**
**HALLMARK_FATTY ACID METABOLISM**	**0.00116195**	**0.009238729**	**HALLMARK_GLYCOLYSIS**	**0.001062**	**0.007135**
HALLMARK_UNFOLDED_PROTEIN_RESPONSE	0.001146789	0.00794155	HALLMARK_INTERFERON_GAMMA_RESPONSE	0.001044	0.007135
HALLMARK_XENOBIOTIC_METABOLISM	0.002107802	0.009238729	HALLMARK_APICAL JUNCTION	0.001089	0.007135
HALLMARK_MTORC1_SIGNALING	0.002162162	0.009238729	**HALLMARK_ADIPOGENESIS**	**0.002079**	**0.010893**
HALLMARK_E2F_TARGETS	0.002202643	0.009238729	**HALLMARK_FATTY ACID METABOLISM**	**0.002179**	**0.010893**
HALLMARK_UV RESPONSE_UP	0.002217295	0.009238729	HALLMARK_ANGIOGENESIS	0.011952	0.037707
**HALLMARK_COAGULATION**	**0.001183432**	**0.00794155**	**HALLMARK_P53_PATHWAY**	**0.003128**	**0.013034**
**HALLMARK_GLICOLYSIS**	**0.004338395**	**0.015494267**	**HALLMARK_MYC_TARGETS_V1**	**0.003093**	**0.013034**
**HALLMARK_HYPOXIA**	**0.00867679**	**0.016287**	**HALLMARK_HYPOXIA**	**0.019958**	**0.049895**
HALLMARK_MYOGENESIS	0.01818188	0.037037037	HALLMARK_ALLOGRAFT_REJECTION	0.012586	0.037707
**HALLMARK_MYC_TARGETS_V2**	**0.016290727**	**0.037024379**			
**HALLMARK_P53_PATHWAY**	**0.023809524**	**0.044091711**			
HALLMARK_PEROXISOME	0.025	0.044642857			

**Table 4 cells-14-01246-t004:** Biological pathways differentially regulated in *HER2*+ tumor basal cell cluster compared to control breast. Bold indicates pathways in common.

Basal Cell Pathways in *Her2*+ Breast Tissue	pval	padj
**HALLMARK_OXIDATIVE_PHOSPHORYLATION**	**0.001059**	**0.006127**
**HALLMARK_INTERFERON_ALPHA_RESPONSE**	**0.001131**	**0.006127**
**HALLMARK_INTERFERON_GAMMA_RESPONSE**	**0.001053**	**0.006127**
HALLMARK_TNFA_SIGNALING_VIA_NFKB	0.001032	0.006127
HALLMARK_APOPTOSIS	0.001089	0.006127
**HALLMARK_COAGULATION**	**0.001144**	**0.006127**
**HALLMARK_P53_PATHWAY**	**0.001042**	**0.006127**
HALLMARK_ALLOGRAFT_REJECTION	0.001115	0.006127
**HALLMARK_ADIPOGENESIS**	**0.001059**	**0.006127**
HALLMARK_UV RESPONSE_UP	0.00216	0.008999
HALLMARK_COMPLEMENT	0.002146	0.008999
**HALLMARK_FATTY_ACID_METABOLISM**	**0.003341**	**0.011136**
HALLMARK_DNA_REPAIR	0.003236	0.011136
HALLMARK_REACTIVE_OXIGEN_SPECIES_PATHWAY	0.001225	0.006127
HALLMARK_INFLAMMATORY_RESPONSE	0.003304	0.011136
HALLMARK_IL6_JAK_STAT3_SIGNALING	0.003695	0.011546
HALLMARK_XENOBIOTIC_METABOLISM	0.013187	0.0333333
HALLMARK_CHOLESTEROL_HOMEOSTASIS	0.008343	0.024539
**HALLMARK_HYPOXIA**	**0.016949**	**0.037481**

**Table 5 cells-14-01246-t005:** Biological pathways in common or differentially regulated in basal and FB2 fibroblast cells in *HER2*+ tumors. Bold indicates pathways in common.

Basal Cell Pathways in *Her2*+ Breast Tissue	pval	padj	FB2 Cell Cluster Pathways in *Her2*+ Breast Tissue	pval	padj
**HALLMARK_OXIDATIVE_PHOSPHORYLATION**	**0.001059**	**0.006127**	**HALLMARK_OXIDATIVE_PHOSPHORYLATION**	**0.00103**	**0.007135**
**HALLMARK_INTERFERON_ALPHA_RESPONSE**	**0.001131**	**0.006127**	HALLMARK_EPITHELIAL_MESENKIMAL_TRANSITION	0.001031	0.007135
**HALLMARK_INTERFERON_GAMMA_RESPONSE**	**0.001053**	**0.006127**	**HALLMARK_INTERFERON_ALPHA_RESPONSE**	**0.001142**	**0.007135**
HALLMARK_TNFA_SIGNALING_VIA_NFKB	0.001032	0.006127	HALLMARK_DNA_REPAIR	0.001063	0.007135
HALLMARK_APOPTOSIS	0.001089	0.006127	**HALLMARK_COAGULATION**	**0.001142**	**0.007135**
**HALLMARK_COAGULATION**	**0.001144**	**0.006127**	HALLMARK_GLYCOLYSIS	0.001062	0.007135
**HALLMARK_P53_PATHWAY**	**0.001042**	**0.006127**	**HALLMARK_INTERFERON_GAMMA_RESPONSE**	**0.001044**	**0.007135**
HALLMARK_ALLOGRAFT_REJECTION	0.001115	0.006127	HALLMARK_APICAL_JUNCTION	0.001089	0.007135
**HALLMARK_ADIPOGENESIS**	**0.001059**	**0.006127**	**HALLMARK_ADIPOGENESIS**	**0.002079**	**0.010893**
HALLMARK_UV RESPONSE_UP	0.00216	0.008999	**HALLMARK_FATTY_ACID_METABOLISM**	**0.002179**	**0.010893**
HALLMARK_COMPLEMENT	0.002146	0.008999	HALLMARK_ANGIOGENESIS	0.011952	0.037707
**HALLMARK_FATTY_ACID_METABOLISM**	**0.003341**	**0.011136**	**HALLMARK_P53_PATHWAY**	**0.003128**	**0.013034**
HALLMARK_DNA_REPAIR	0.003236	0.011136	HALLMARK_MYC_TARGETS	0.003093	0.013034
HALLMARK_REACTIVE_OXIGEN_SPECIES_PATHWAY	0.001225	0.006127	**HALLMARK_HYPOXIA**	**0.019958**	**0.049896**
HALLMARK_INFLAMMATORY_RESPONSE	0.003304	0.011136	HALLMARK_ALLOGRAFT_REJECTION	0.012588	0.037707
HALLMARK_IL6_JAK_STAT3_SIGNALING	0.003695	0.011546			
HALLMARK_XENOBIOTIC_METABOLISM	0.013187	0.0333333			
HALLMARK_CHOLESTEROL_HOMEOSTASIS	0.008343	0.024539			
**HALLMARK_HYPOXIA**	**0.016949**	**0.037481**			

**Table 6 cells-14-01246-t006:** Biological pathways in common or differentially regulated in FB2 fibroblast and uncharacterized UNC1 cell cluster in *HER2*+ tumors and control breast. Bold indicates pathways in common.

FB2 Fibroblast Pathways in *Her2*+ Breast Tissue	pval	padj	UNC1 Cell Cluster Pathways in *Her2*+ Breast Tissue	pval	padj
**HALLMARK_OXIDATIVE_PHOSPHORYLATION**	**0.00103**	**0.007135**	**HALLMARK_OXIDATIVE_PHOSPHORYLATION**	**0.001148**	**0.012903**
**HALLMARK_EPITHELIAL_MESENCHYMAL_TRANSITION**	**0.001031**	**0.007135**	**HALLMARK_INTERFERON GAMMA RESPONSE**	**0.001183**	**0.012903**
**HALLMARK_INTERFERON_ALPHA_RESPONSE**	**0.001142**	**0.007135**	**HALLMARK_HYPOXIA**	0.001229	0.012903
**HALLMARK_DNA_REPAIR**	**0.001063**	**0.007135**	**HALLMARK_ALLOGRAFT_REJECTION**	0.01255	0.012903
HALLMARK_COAGULATION	0.01142	0.007135	**HALLMARK_INTERFERON_ALPHA_RESPONSE**	**0.00129**	**0.012903**
**HALLMARK_GLYCOLYSIS**	**0.001062**	**0.007135**	**HALLMARK_MYC_TARGETS**	0.002222	0.015152
**HALLMARK_INTERFERON_GAMMA_RESPONSE**	**0.001044**	**0.007135**	**HALLMARK_P53_PATHWAY**	0.002424	0.015152
HALLMARK_APICAL JUNCTION	0.001089	0.007135	HALLMARK_TNFA_SIGNALING_VIA_NFKB	0.002317	0.015152
HALLMARK_ADIPOGENESIS	0.002079	0.010893	HALLMARK_DNA_REPAIR	0.004957	0.027793
HALLMARK_FATTY ACID METABOLISM	0.002179	0.010893	HALLMARK_REACTIVE_OXIGEN_SPECIES_PATHWAY	0.007267	0.027793
HALLMARK_ANGIOGENESIS	0.011952	0.037707	**HALLMARK_GLYCOLYSIS**	**0.007782**	**0.027793**
**HALLMARK_P53_PATHWAY**	**0.003128**	**0.013034**	HALLMARK_INFLAMMATORY_RESPONSE	0.013263	0.009384
**HALLMARK_MYC_TARGETS_V1**	**0.003093**	**0.013034**	HALLMARK_KRAS_SIGNALING_UP	0.013459	0.039585
**HALLMARK_HYPOXIA**	**0.019958**	**0.049895**	HALLMARK_IL6_JAK_STAT3_SIGNALING	0.012931	0.039585
**HALLMARK_ALLOGRAFT_REJECTION**	**0.012586**	**0.037707**			

**Table 7 cells-14-01246-t007:** Biological pathways differentially regulated in basal, plasma, and natural killer cell clusters in HER2+ tumors compared to control breast. Bold indicates pathways in common.

B Cell Cluster Pathway in *Her2*+ Breast Tissue	pval	padj	Natural Killer Cell Cluster Pathway in *Her2*+ Breast Tissue	pval	padj
**HALLMARK_INTERFERON_GAMMA_RESPONSE**	**0.001182**	**0.009411**	**HALLMARK_ALLOGRAFT_REJECTION**	**0.01189**	**0.005746**
**HALLMARK_TNFA_SIGNALING_VIA_NFKB**	0.001135	0.009411	**HALLMARK_INTERFERON GAMMA RESPONSE**	**0.001193**	**0.005746**
**HALLMARK_ALLOGRAFT_REJECTION**	**0.01202**	**0.009411**	**HALLMARK_TNFA_SIGNALING_VIA_NFKB**	**0.001155**	**0.005746**
**HALLMARK_INTERFERON_ALPHA_RESPONSE**	0.001318	0.009411	**HALLMARK_INTERFERON_ALPHA_RESPONSE**	**0.001264**	**0.005746**
**HALLMARK_UV_RESPONSE_UP**	0.001252	0.009411	**HALLMARK_UV_RESPONSE_UP**	0.001233	0.005746
**HALLMARK_MTORC1_SIGNALING**	0.001145	0.009411	**HALLMARK_IL2_STAT5_SIGNALING**	**0.001192**	**0.005746**
**HALLMARK_HYPOXIA**	**0.001186**	**0.009411**	**HALLMARK_MTORC1_SIGNALING**	0.001166	0.005746
**HALLMARK_INFLAMMATORY_RESPONSE**	**0.002635**	**0.016469**	**HALLMARK_APOPTOSIS**	**0.001193**	**0.005746**
HALLMARK_FATTY_ACID_METABOLISM	0.004087	0.022707	**HALLMARK_INFLAMMATORY_RESPONSE**	**0.001252**	**0.005746**
**HALLMARK_APOPTOSIS**	**0.007151**	**0.034722**	HALLMARK_OXIDATIVE_PHOSPHORYLATION	0.001193	0.005746
**HALLMARK_IL2_STAT5_SIGNALING**	**0.008235**	**0.034722**	**HALLMARK_HYPOXIA**	**0.001221**	**0.005746**
**Plasma cell cluster pathway in *Her2*+ breast tissue**	**pval**	**padj**	HALLMARK_COMPLEMENT	0.004969	0.019111
**HALLMARK_INTERFERON_ALPHA_RESPONSE**	**0.00137**	**0.034247**	HALLMARK_P53_PATHWAY	0.005967	0.020704
**HALLMARK_ALLOGRAFT_REJECTION**	**0.001252**	**0.034247**	HALLMARK_IL6_JAK_STAT3_SIGNALING	0.015048	0.03762
			HALLMARK_BILE_ACID_METABOLISM	0.01385	0.036448

## Data Availability

scRNA-seq data from publicly available datasets of adult human breast tissue were downloaded from Gene Expression Omnibus (GEO) repository: (https://www.ncbi.nlm.nih.gov/geo/query/acc.cgi?acc=GSE195665) (accessed 20 July 2025) for adult normal healthy breast and (https://www.ncbi.nlm.nih.gov/geo/query/acc.cgi?acc=GSE176078) (accessed 20 July 2025) for human breast cancer patients [19,20]. Five normal breast samples (GSM7500385hbcac76, GSM7500386hbcac77, GSM7500417hbcac108, GSM7500426hbcac117, GSM7500458hbcac149) and 5 HER2+ samples (CID3586, CID3838, CID3921, CID4066, CID45171) were selected for this analysis from 126 samples collected from MD Anderson (MDA), UC Irvine (UCI) or Baylor College of Medicine (BCM). In addition to the clinical metadata, adult normal breast samples were selected based on the procedure from which the tissue source was collected.

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
