# Peer review of "Single-Cell mRNA Analysis for the Identification of Molecular Pathways of IRF1 in HER2+ Breast Cancer"

_cells, 2025, doi:10.3390/cells14161246_

Round 1

Reviewer 1 Report

Comments and Suggestions for Authors

The article presents an original and valuable approach to understanding the pleiotropic roles of IRF1 in HER2+ breast cancer at the single-cell level. The paper contains a large amount of data and interesting comparisons of gene expression, but requires significant methodological, editorial and structural improvements for greater clarity and scientific precision.

Comments and Suggestions for Authors:
1. Clarity of hypothesis and goals:
The authors present multiple goals (role of IRF1, RNASET2, TMEM230, SDC2), but they are not clearly stated in the introduction. It is recommended to shorten and clarify the main research hypothesis.

2. Complexity of the text:
The paper contains excessively extensive paragraphs, often mixing experimental data with the literature review. It is advisable to organize and shorten them.

3. Methods:
scRNA-seq analyses and use of GEO data are thoroughly described.
However, there is a lack of information on the bioinformatics tools used, data quality metrics, and details of dataset integration.

4. Results and interpretation:
Observations of IRF1 expression in different cell clusters are interesting and potentially significant.

Conclusions regarding the function of IRF1 as a tumor promoter or suppressor are too generalized – they should be better embedded in quantitative data and pathway analysis.

5. Lack of experimental validation:
The work is based almost exclusively on transcriptomic and gene expression data. There is a lack of confirmation, e.g. immunofluorescence, Western blot, or protein analysis.

6. Language and structure:

The style is sometimes unreadable, with repetitions and colloquialisms (e.g. “designer isoforms”, “rogue acting tumor cells”). Language editing needed.

Furthermore, I rate the work highly in terms of science. After the above-mentioned corrections, the work can be accepted for publication in the journal Cells.

Comments on the Quality of English Language

The manuscript is written in English that is generally understandable, but requires substantial revision for clarity, grammar, and fluency. I recommend proofreading by a Native Speaker.

Author Response

The authors want to sincerely thank this reviewer for generously providing helpful comments and suggestions for improving the manuscript and our research study. We apologize for the late revisions and return comments from us. In part this was due to a tragic loss of one of the authors who contributed the primary bioinformatics analysis. We apologize for this delay.

The article presents an original and valuable approach to understanding the pleiotropic roles of IRF1 in HER2+ breast cancer at the single-cell level. The paper contains a large amount of data and interesting comparisons of gene expression, but requires significant methodological, editorial and structural improvements for greater clarity and scientific precision.

Comments and Suggestions for Authors:
1. Clarity of hypothesis and goals:
The authors present multiple goals (role of IRF1, RNASET2, TMEM230, SDC2), but they are not clearly stated in the introduction. It is recommended to shorten and clarify the main research hypothesis.

We have taken the reviewer’s comments and suggestions and have shortened the manuscript and extensively revised most of the manuscript in order to promote clarity and brevity. We hope that the revisions of the Introduction and Abstract allow for a better understanding of our goals and hypotheses.

2. Complexity of the text:
The paper contains excessively extensive paragraphs, often mixing experimental data with the literature review. It is advisable to organize and shorten them.

We have shorted the manuscript and removed and combined most of the paragraphs. To prevent confusion of experimental data from our previous studies and the literature we have moved the previous published data to the Introduction and have left salient information to allow for the reader to understand our motivation in the design and implementation of the new experiments shown in the Results section. 

3. Methods:
scRNA-seq analyses and use of GEO data are thoroughly described.
However, there is a lack of information on the bioinformatics tools used, data quality metrics, and details of dataset integration.

We apologize we have not provided more extensive information on our bioinformatics approach, part of this was due to relying on  references our previous research on TMEM230 using RA patients and the passing away of one of the authors who contributed primarily to the bioinformatics analysis.. We have therefore added the requested information and description of the data requested by this reviewer and have provided additionally information concerning patients also in the Supplementary Materials section. 

4. Results and interpretation:
Observations of IRF1 expression in different cell clusters are interesting and potentially significant.

Conclusions regarding the function of IRF1 as a tumor promoter or suppressor are too generalized – they should be better embedded in quantitative data and pathway analysis.

We have tried to be more specific with the potential functions attributed to IRF1 by our expression analysis, and have rewritten this part to correlate better the overlapping functions of the previous gene pathways we have identified in RA with pathways uncovered in cancer.

5. Lack of experimental validation:
The work is based almost exclusively on transcriptomic and gene expression data. There is a lack of confirmation, e.g. immunofluorescence, Western blot, or protein analysis.

We apologize if it seemed that were was lack of cell and in vitro analysis performed with the human patient derived cells.  The cell and in vitro analysis was relegated to the Supplementary Material at the advice of the editor as there were too many other figures. We have added the Supplementary Material  data at the end of the manuscript that indicates extensive tissue culture work and validation using IRF1 lentiviral based inhibition of active assays.

6. Language and structure:

The style is sometimes unreadable, with repetitions and colloquialisms (e.g. “designer isoforms”, “rogue acting tumor cells”). Language editing needed.

We apologize for the repetitions and difficulty in reading the manuscript. The manuscript has been extensively revised to remove grammatical and English syntax errors. Terminology  not commonly recognized  such as designer isoforms”, “rogue acting tumor cells” were removed.

We again want to thank the reviewer for their time for helping us improve the manuscript and experimental study. The comments and suggestions were most courteous.

Reviewer 2 Report

Comments and Suggestions for Authors

The manuscript by Vilardo and colleagues studied the opposing roles of IRF1 in breast cancer. Specifically, they utilized single-cell RNA-seq and bioinformatics analysis and revealed that IRF1 and RNASET2 were differentially expressed between normal and tumor tissue cells, and co-regulated in HER2+ tumors based on functional analysis. Thus, they proposed IRF1 and RNASET2 as potential targets for breast cancer treatment. 

The major bottleneck of the current manuscript lies in twofold. (1) The logic flow needs to be much improved as in the current version the various analyses are disintegrated, making it hard for readers to follow the motivation. (2) The single-cell transcriptomics analysis and results presentation need to be revised. Various issues in the differential analysis and the cell cluster specificity of IRF1 need to be addressed. The tables (Table 1-7) should be presented as real tables, instead of pictures. 

Comments and suggestions:
1.    Line 46-107, the authors made good background relating to their previous research findings. However, it would be better if the authors could present it into a more incremental way. That is, what was found in the old research, what was lacking and what this one brings in to fill the gap. 
2.    Line 181-199, the authors should provide more details regarding the data subset and processing. For example, GSE195665 contains data from three platforms, Visium, scRNA-seq and snRNA-seq. Which one was used in this study? When comparing the normal and tumor tissue cells, what normalization method was used to overcome the batch effects of multiple data sources? “In addition to the clinical metadata, adult normal breast samples were selected…” (Line 190-191), for the sake of clarity, it would be better that the authors make a list of the accession numbers of selected samples as part of the Supplementary Table. 
3.    Line 202-210, this is confusing—is the synovial tissue result (Figure 1) obtained from the previous study or the current one (i.e. the single-cell RNA-seq data analysis)?  It seems that neither GSE195665 nor GSE176078 included RA and OA patients. 
4.    Line 231-232, what does “designer” protein isoforms mean? The sentence appears truncated… 
5.    Line 345-348, Figure 4, the authors are recommended to present IRF1’s expression profiles contrasting HER2+ and Control in each cell type of interest as boxplots or vioplots. The current UMAP plot cannot show the cell cluster identity. Also, the difference coded in the color between two groups is visually hard to quantify. 
6.    Figure 5, from the y-axis range (1 to 5), IRF1 doesn’t count as the “highly expressed”. Also, the conclusion about its specificity in UNC2, FB2, FB1, and the SM cell cluster is questionable as almost all cell clusters express it (except UNC3/4 and EPl1). 

Author Response

The authors want to sincerely thank this reviewer for generously providing helpful comments and suggestions for improving the manuscript and our research study. We apologize for the late revisions and return comments from us. In part this was due to a tragic loss of one of the authors who contributed the primary bioinformatics analysis. 

The manuscript by Vilardo and colleagues studied the opposing roles of IRF1 in breast cancer. Specifically, they utilized single-cell RNA-seq and bioinformatics analysis and revealed that IRF1 and RNASET2 were differentially expressed between normal and tumor tissue cells, and co-regulated in HER2+ tumors based on functional analysis. Thus, they proposed IRF1 and RNASET2 as potential targets for breast cancer treatment. 

The major bottleneck of the current manuscript lies in twofold. (1) The logic flow needs to be much improved as in the current version the various analyses are disintegrated, making it hard for readers to follow the motivation.

We have extensively revised the manuscript to improve clarity and motivation of our goals. For instance, Abstract and Introduction was rewritten with this in mind. 

Various issues in the differential analysis and the cell cluster specificity of IRF1 need to be addressed. The tables (Table 1-7) should be presented as real tables, instead of pictures. 

We thank the reviewer for this important comment. 

Tables 1-7 will be submitted to the journal as the journal rules require tables to be Excel files. In the manuscript images were added as a convenience to improve formatting within the text.

Comments and suggestions:
1.    Line 46-107, the authors made good background relating to their previous research findings. However, it would be better if the authors could present it into a more incremental way. That is, what was found in the old research, what was lacking and what this one brings in to fill the gap. 

We thank the reviewer for this important suggestion and we therefore have rewritten the Introduction to relay in an incremental way our previous results and experimental data. We hope the rewriting help clarify what was previously identified by our research and the new experimental data.

2.    Line 181-199, the authors should provide more details regarding the data subset and processing. For example, GSE195665 contains data from three platforms, Visium, scRNA-seq and snRNA-seq. Which one was used in this study? When comparing the normal and tumor tissue cells, what normalization method was used to overcome the batch effects of multiple data sources? “In addition to the clinical metadata, adult normal breast samples were selected…” (Line 190-191),

We have provided additional information regarding the accession of the data sets and their processing. We apologize for not providing more information in the beginning. 

for the sake of clarity, it would be better that the authors make a list of the accession numbers of selected samples as part of the Supplementary Table. 

We agree with adding accession numbers and description of the patient data, and this was added to the Supplementary Material.

3.    Line 202-210, this is confusing—is the synovial tissue result (Figure 1) obtained from the previous study or the current one (i.e. the single-cell RNA-seq data analysis)?  It seems that neither GSE195665 nor GSE176078 included RA and OA patients. 

We apologize for this confusion. We have previously identified IRF1 in RA patients but have not reported previously. We therefore rewritten the relevant section to make it clear that this is new information. We have added the RA and OA accession data to the Materials and Methods. 

4.    Line 231-232, what does “designer” protein isoforms mean? The sentence appears truncated…

We apologize, the sentence containing the terminology has been removed and part of this paragraph has been removed to provide greater clarity.

5.    Line 345-348, Figure 4, the authors are recommended to present IRF1’s expression profiles contrasting HER2+ and Control in each cell type of interest as boxplots or vioplots. The current UMAP plot cannot show the cell cluster identity. 

We agree that the UMAP plot cannot provide information concerning the cell cluster identify. We therefore show for each gene analyzed such as RNASET2 and SDC2 for each cell cluster type in following in box plots. For instance, Figure 6 corresponds to Figure 5 for IRF1, and Figure 7 has a corresponding box plot shown in Figure 9 for SDC2, and Figure 11 is the corresponding box plot for RNASET2.

6.    Figure 5, from the y-axis range (1 to 5), IRF1 doesn’t count as the “highly expressed”. 

We thank the reviewer for this observation and have revised the sentence. Other genes involved in cell signaling we have studied were shown to have even lower expressions but likely have biological roles in cells. For instance, rarely have we found transcription factors to be highly expressed.

Also, the conclusion about its specificity in UNC2, FB2, FB1, and the SM cell cluster is questionable as almost all cell clusters express it (except UNC3/4 and EPl1).

We agree and have revised the manuscript to not promote this confusion. We meant to indicate that IRF1 and the other genes analyzed often have specific expression levels in the different cell types.

We like to thank the reviewer again for their generous time in helping us improve the manuscript and clarity of the study. The reviewer was most courteous.

Round 2

Reviewer 1 Report

Comments and Suggestions for Authors

Significant Improvement Noted: The revised version demonstrates substantial efforts to clarify the main research objectives and better organize the experimental framework. The addition of well-labeled figures and more structured methodology enhances readability and interpretability.

While the study is methodologically sound and the revised version shows significant improvements, the manuscript still needs focused language editing, clearer narrative structure, and a more concise presentation of results and conclusions. Addressing these issues will strengthen the manuscript’s impact and readability.

Comments on the Quality of English Language

A thorough professional language editing is recommended. Ideally, the manuscript should be reviewed by a native English-speaking editor with scientific expertise to ensure clarity, precision, and readability throughout.

Author Response

We want to thank the reviewer again for their generosity in reviewing our study and helping in improving the language style. Keeping in mind of the suggestions and comments presented by this reviewer, the manuscript was revised completely with additional feed back from outside  experts in the field of the study. As the corresponding author's native language is American english, outside experts were chosen to revise the manuscript who speak european english. Additionally, the editorial department of the journal has kindly offered to assist in the final editorial proofing upon acceptance of the manuscript for publication.

We want to thank the review most kindly for their time and important suggestions and comments in helping us produce the best manuscript possible for publication.

We thank you again.

Sincerely, the authors.

Reviewer 2 Report

Comments and Suggestions for Authors

It is very sad to hear the passing of Dr. Edoardo Abeni. The authors’ responses have addressed most concerns and I do not have other questions. There is one minor issue—the gene enrichment results in Table 1-7 could have had the “fold of enrichment” (namely, the odds ratio) or the number of hits besides the p-value. Typically, an enrichment analysis reports both the “size effect” (as the odds ratio or the number of hits), and the statistical significance (as the p-value). However, this is nothing critical and the authors may keep it as is if they prefer.

Author Response

Dear Reviewer

We very much thank you for you converting your sympathy in regards to the passing away of our Edoardo. We want to thank you again for helping us improve the manuscript with your suggestions and comments. We could not be happier in how the manuscript turned out with your assistance.

thank you again